A cross-ocean comparison of responses to settlement cues in reef-building corals

Davies Sarah W. 1 daviessw@gmail.com
Meyer Eli 2
Guermond Sarah M. 2
Matz Mikhail V. 1
1 Department of Integrative Biology, University of Texas at Austin , Austin, TX , United States
2 Department of Integrative Biology, Oregon State University , Corvallis, OR , United States
Qian Pei-Yuan
Electronic publication date: 2014 Apr 8
Publication date: 2014
Volume: 2
Electronic Location ID: e333
Received 2013 Dec 30; Accepted 2014 Mar 15
Copyright: © 2014 Davies et al.
Copyright year: 2014
Copyright holder: Davies et al.
License: This is an open access article distributed under the terms of the Creative Commons Attribution License, which permits unrestricted use, distribution, and reproduction in any medium, provided the original author and source are credited.
License URL: https://creativecommons.org/licenses/by/3.0/

Keywords: 18S rRNA, Coral recruitment, Crustose coralline algae, Settlement cues, Meta-barcoding, OTU

Funding: National Science Foundation DEB-1054766 Research was funded by the National Science Foundation grant DEB-1054766 to MVM, a departmental start-up grant from the Section of Integrative Biology at the University of Texas at Austin to SWD and the PADI Foundation Award to SWD. The funders had no role in study design, data collection and analysis, decision to publish, or preparation of the manuscript.

==============================
Caribbean coral reefs have deteriorated substantially over the past 30 years, which is broadly attributable to the effects of global climate change. In the same time, Indo-Pacific reefs maintain higher coral cover and typically recover rapidly after disturbances. This difference in reef resilience is largely due to much higher coral recruitment rates in the Pacific. We hypothesized that the lack of Caribbean recruitment might be explained by diminishing quality of settlement cues and/or impaired sensitivity of Caribbean coral larvae to those cues, relative to the Pacific. To evaluate this hypothesis, we assembled a collection of bulk samples of reef encrusting communities, mostly consisting of crustose coralline algae (CCA), from various reefs around the world and tested them as settlement cues for several coral species originating from different ocean provinces. Cue samples were meta-barcoded to evaluate their taxonomic diversity. We observed no systematic differences either in cue potency or in strength of larval responses depending on the ocean province, and no preference of coral larvae towards cues from the same ocean. Instead, we detected significant differences in cue preferences among coral species, even for corals originating from the same reef. We conclude that the region-wide disruption of the settlement process is unlikely to be the major cause of Caribbean reef loss. However, due to their high sensitivity to the effects of climate change, shifts in the composition of CCA-associated communities, combined with pronounced differences in cue preferences among coral species, could substantially influence future coral community structure.

Introduction

The majority of reef-building corals are broadcast-spawning species that release gametes annually to produce planktonic larvae that are dispersed by ocean currents (Baird, Guest & Willis, 2009). Reef recovery after disturbances, such as catastrophic bleaching events or hurricanes, is critically dependent on the successful recruitment of these planktonic larvae back to reefs (Buston et al., 2012). Coral reefs worldwide are declining at accelerating rates, which has been generally attributed to the increase in both global and local anthropogenic stressors (Hoegh-Guldberg et al., 2007). The specific factors driving this decline, including those affecting coral recruitment, are the subject of active ongoing research.

While coral cover has been declining in Indo-Pacific reefs in recent years (Bruno & Selig, 2007; Wakeford, Done & Johnson, 2008; De’ath et al., 2012), their higher biodiversity and range of recruitment and post-recruitment strategies appear to make these reefs more resilient (Adjeroud et al., 2009; Roff & Mumby, 2012). Caribbean reefs exhibit lower resilience than Indo-Pacific reefs, which has been attributed to several factors including recruitment failure (Connell, Hughes & Wallace, 1997; Roff & Mumby, 2012). Across the Caribbean, recruitment rates of broadcast spawning corals are consistently low (Hughes & Tanner, 2000; Gardner et al., 2003; Vermeij, 2006; Davies, Matz & Vize, 2013), even though large reef builders still dominate coral cover on Caribbean reefs (Kramer, 2003). Instead, brooding genera such as Agaricia and Porites are the dominant coral species recruiting on Caribbean reefs (Bak & Engel, 1979; Green, Edmunds & Carpenter, 2008; Davies, Matz & Vize, 2013). Spectacular recoveries after disturbances are not uncommon on Pacific reefs (i.e., Golbuu et al., 2007), but comparable levels of recovery have not been documented in the Caribbean (but see Carpenter & Edmunds, 2006; Idjadi et al., 2006). A comparative study of proximal causes of this difference in coral recruitment among ocean regions could elucidate some of the main drivers of Caribbean recruitment failure.

In principle, low recruitment rates might result from a variety of factors such as reduced coral population sizes, poor spawning synchrony, low fertilization rate, or high mortality (either pre- or post-settlement). Some of these potential explanations are unlikely to apply to the Caribbean-wide recruitment failure. For example, adult population sizes, at least for some Caribbean reefs, are still adequate and spawning remains highly synchronous and prolific (i.e., Flower Garden Banks, Vize et al., 2005). High fertilization success is also observed under natural conditions (Levitan et al., 2004). While pre- and post-settlement mortality remains among the main potential causes, it is also possible that the effects of climate change in the Caribbean may have disrupted ecological interactions required for the recruitment process itself (Harrison, 1990), specifically the interaction between coral larvae and natural settlement cues.

Various factors influence coral settlement (Maida, Coll & Sammarco, 1994; Mundy & Babcock, 1998; Raimondi & Morse, 2000), however for many corals the biological properties of the reef surface appear to play a pivotal role in this choice (Babcock & Mundy, 1996; Heyward & Negri, 1999; Price, 2010; Ritson-Williams et al., 2010). Crustose coralline algae (CCA; Rhodophyta, Corallinaceae) and associated communities have been shown to be one of the primary inducers of settlement and metamorphosis in coral larvae (Morse & Morse, 1988; Morse et al., 1996; Heyward & Negri, 1999). While marine bacteria also influence settlement in coral larvae (Negri et al., 2001; Tebben et al., 2011; Tran & Hadfield, 2011), recent work demonstrates that CCA species known to elicit the strongest settlement responses are also the most affected by the changes in ocean chemistry associated with climate change (Anthony et al., 2008; Doropoulos et al., 2012; Smith et al., 2013), suggesting that changes in these CCA communities might be responsible for reduced coral recruitment.

We hypothesized that the correspondence between coral larval preferences and availability/quality of settlement cues (CCA associated communities) on Caribbean reefs may have broken down, resulting in reduced coral recruitment. This mismatch may take two forms: (1) appropriate settlement cues may be present, but larvae have lost the ability to respond to them, or (2) larval responses remain intact, but effective settlement cues are absent. To evaluate these possibilities, we performed reciprocal preference trials for three species of broadcast spawning Caribbean corals (Montastraea franksi, Diploria strigosa and Stephanocoenia intersepta) and four Indo-Pacific corals (Acropora millepora, Acropora tenuis, Favia lizardensis and Ctenactis echinata). Larval response of each species was tested against a collection of seven samples of CCA-associated communities from various locations in the Caribbean (n = 3) and the Indo-Pacific (n = 4). Since we were not interested in characterizing larval responses to particular CCA species but rather wanted to generally evaluate cue presence-absence in the environment, we collected whole encrusting communities from reef top or rubble to better approximate what coral larvae might encounter in nature rather than picking specific CCA species. To evaluate the diversity of the cues tested, their taxonomic composition was characterized post hoc by metabarcoding based on the eukaryotic ribosomal 18S rRNA gene.

Materials and Methods

Settlement cue collections

Collections of CCA associated communities (which we will refer to as “cue ∗ s” from now on) from a number of locations in the Caribbean and Pacific was assembled (Table 1). Caribbean locations included the Florida Keys (FF), the Flower Garden Banks (FGB) and Bonaire (B). Pacific locations included Orpheus Island (Great Barrier Reef, Australia: A1, A2), Pohnpei (P) and Guam (G). Samples were stored in seawater at 80 °C.

Table 1 Settlement cue panel and metabarcoding statistics.

CCA cue information including: name of the cue, site the cue was collected at and the oceanographic region the site was located in. Metabarcoding statistics including: number of quality-filtered reads, number of operational taxonomic units (OTUs), number of reads uniquely mapping to OTUs and the mapping efficiency of the reads.

Cue	Site	Region	# of quality-
filtered reads	# of OTUs	# of reads uniquely
mapping to OTUs	Mapping efficiency	
A1	Orpheus Island (GBR)	Pacific	2760	6	2714	0.983	
A2	Orpheus Island (GBR)	Pacific	4906	10	3566	0.727	
B	Bonaire	Caribbean	1447	8	1222	0.844	
FF	Florida Keys	Caribbean	2762	10	2411	0.873	
FGB	Flower Garden Banks	Caribbean	2492	9	2341	0.939	
G	Guam	Pacific	4495	11	2963	0.659	
P	Pohnpei	Pacific	NA	NA	NA	NA	

Caribbean spawn I

On the evening of August 31, 2010 (eight days after the full moon), during the annual coral spawning event at the Flower Garden Banks National Marine Sanctuary (FGBNMS), gamete bundles were collected with mesh nets directly from three distinct Montastraea (Orbicella) franksi colonies. Bundles were brought to the surface, cross-fertilized for one hour and then excess sperm was removed by rinsing through 150 µm nylon mesh. Larvae were reared in 1 µm filtered seawater (FSW) in three replicate plastic culture vessels at 5 larvae per ml. Larvae were transferred to the laboratory at the University of Texas at Austin on September 1, 2010. Samples were collected under the FGBNMS permit # FGBNMS-2009-005-A2.

Preliminary competency experiments assayed with several CCA samples determined that M. franksi larvae did not reach competence until 14 days post-fertilization, therefore CCA preference trials were started at this age. To quantify the responsiveness of settlement-competent larvae to six different cue samples (Table 1), twenty larvae per well were transferred into 10 ml of FSW in 6-well plates. Cue samples were finely ground with a mortar and pestle shortly before the settlement trials and a single drop of the resulting uniform slurry was added to each well (n = 4 well replicates per cue, randomly assigning cues to wells). Four FSW control treatments were also included. The proportion of metamorphosed larvae (visual presence of septa) was quantified after 48 h using a fluorescent stereomicroscope MZ-FL-III (Leica, Bannockburn, IL, USA) equipped with F/R double-bandpass filter (Chroma no. 51004v2) (Figs. 1B and 1C).

Figure 1 Settlement responses of M. franksi from the Flower Garden Banks in 2010.

Settlement responses of M. franksi from the Flower Garden Banks in 2010 (mean ± SE). (A) Proportion of coral settlement. Darker bars correspond to Caribbean cues, lighter bars to Pacific cues. (B) Fluorescent photograph of M. franksi larvae before settlement. (C) Fluorescent photograph of M. franksi recruit post-settlement.

Pacific spawn I

In November 2010, at Orpheus Island Research Station, Great Barrier Reef, Australia, the same type of experiments as described in the previous section were conducted with the same panel of cues (plus an additional Australian cue, A2). Four species of broadcast spawning corals were tested: Acropora millepora, A. tenuis, Favia lizardensis, and Ctenactis echinata. Adult corals were collected and maintained in raceways until spawning at which point they were isolated in 20-gallon plastic bins. Following spawning, gametes were collected from several colonies and cross-fertilized as described above. Initial trials to test for larval competency were conducted and final data were collected on 5d-old larvae, although C. echinata were never observed to settle over a period of several weeks, even in response to GLWamide (data not shown). Settlement assays were conducted as in the 2010 Caribbean Spawn I described above, the only differences being inclusion of A2 cue and increase of per-cue replication level to n = 6 (Table 1). Samples for Australian fieldwork were collected under Great Barrier Reef Marine Park Authority permit number G10/33943.1.

Caribbean spawn II

On the evening of August 18, 2011 (eight days after the full moon), gamete bundles from multiple colonies of three broadcast-spawning Caribbean coral species were collected from FGBNMS (Diploria strigosa, Montastraea franksi & Stephanocoenia intersepta). Gametes were cross-fertilized and maintained in similar conditions as in 2010 and transferred to the laboratory at the University of Texas at Austin on August 21, 2011. Samples were collected under permit FGBNMS-2009-005-A3. Settlement assays were conducted on all species across all cues in the panel including A2 (n = 6 per cue). D. strigosa trials were conducted on four day old larvae after initial testing for competence and M. franksi trials were completed at 21 days old after competence was determined. S. intersepta were never observed to settle over a period of two months.

Metabarcoding of cue communities

In order to determine the taxonomic composition of each cue sample, we used deep amplicon sequencing. DNA was isolated from ground-up cue samples as described in Davies et al. (2013). The conserved 5′ portion of the eukaryotic small-subunit ribosomal RNA gene (18S SSU) was amplified via PCR using the SP-F-30 forward primer (5′ TCTCAAAGACTAAGCCATGC 3′) and the reverse primer SP-R-540 (5′ TTACAGAGCTGGAATTACCG 3′) (Vidal, Meneses & Smith, 2002). Each 30 µl polymerase chain reaction (PCR) mixture contained 10 ng of DNA template, 0.1 µM forward primer, 0.1 µM reverse primer, 0.2 mM dNTP, 3 µl 10X ExTaq buffer, 0.025 U ExTaq Polymerase (Takara Biotechnology) and 0.0125 U Pfu Polymerase (Agilent Technologies), and was amplified using a DNA Engine Tetrad2 Thermal Cycler (Bio-Rad, Hercules, CA, USA) with a cycling profile of 94 °C 5 min−(94 °C 40 s−55 °C 2 min−72 °C 60 s) × N−72 °C 10 min, with N = 17–24 depending on the sample. Amplicons (∼550 bp bands) were successfully obtained from 6 out of 7 samples (Pohnpei sample failed to amplify despite increased cycle numbers and repeated attempts). Amplicons were cleaned using PCR clean-up kit (Fermentas), 10 ng of the cleaned product was used as template in a second PCR to incorporate 454-Titanium primers and unique barcodes. Each PCR contained 0.1 µM of the universal Btn-SPR-F forward primer (5′ CCTATCCCCTGTGTGCCTTGGCAGTCTCAGTCTCAAAGACTAAGCCATGC 3′, underlined stretch matches SP-F-30 primer) and 0.1 µM of unique reverse primer containing a 4-bp barcode (5′ CCATCTCATCCCTGCGTGTCTCCGACTCAGTACTTTACAGAGCTGGAATTACCG 3′, underlined stretch matches SP-R-540 primer, bold indicates 4 bp barcode). The cycling profile was 95 °C 5 min−(95 °C 30 s−55 °C 30 s−72 °C 60 s) × 4−72 °C 5 min. Amplicons were gel-purified and pyrosequenced using 454-FLX (Roche) with Titanium chemistry at the Genome Sequencing and Analysis Facility (GSAF) at the University of Texas at Austin. All cue samples were sequenced with the exception of Pohnpei, which we were unable to amplify, even with additional efforts involving modifying DNA template concentration and PCR cycle numbers.

Resulting reads were split by barcode and trimmed using a custom Perl script that removes adaptors, barcodes and low quality read ends. Reads that became shorter than 250 bp after this trimming step were discarded. Reads were then clustered at 97% identity using the program cd-hit-454 (Huang et al., 2010). The longest sequences from clusters containing >1% of the filtered reads were selected as representatives of distinct operational taxonomic units (OTUs) and used as reference sequences for mapping the filtered reads using the runMapping module of Newbler v. 2.6 (Roche) with repeat score threshold (parameter –rst) of 3 (i.e., a read was considered uniquely mapped if its best hit among OTU sequences was different from the next-best hit by 3 or more additionally aligned bases). The proportion of reads uniquely mapping to a particular OTU was taken as a measure of the relative abundance of this OTU in the sample. All OTUs accounting for ≥ 1% mapped reads were assigned to their most likely taxonomic order based on BLAST matches (Altschul et al., 1997) against nonredundant (nr) NCBI database. The non-metric multidimensional scaling (NMDS) analysis based on Bray-Curtis similarities of relative proportions of observed orders was performed using the vegan package in R (Jari Oksanen et al., 2013).

To evaluate the degree to which our sequencing coverage captured sequence diversity in each sample, we conducted rarefaction analysis. The reads mapping to major OTUs (OTUs comprising ≥ 1% of each sample) were randomly resampled at various depths to simulate the effects of lower sequencing coverage. For each simulated sequencing depth, we randomly sampled with replacement and counted the number of OTUs identified in the sampled subset. Sampling was performed 1000 times for each simulated sequencing depth to calculate the average number of OTUs detected at each depth (Fig. S1). Perl script for rarefaction analysis (cca_rarefaction.pl) and R script for plotting rarefaction curves (rarefaction_figs.R) are available in Supplemental Information 1.

To further characterize the taxonomic diversity of cue samples, two OTUs accounting for the highest proportion of reads within each sample (together representing 39.4–68.3% of the total mapped reads in a cue sample) (Table 2) were aligned using MAFFT version 7 (Katoh & Standley, 2013). This alignment was then used to construct a neighbor-joining tree in BIONJ (Gascuel, 1997) with 1000 bootstrap replicates. This tree was downloaded in Newick format and modified for visualization using FigTree V1.4.0 (http://tree.bio.ed.ac.uk/software/figtree/).

Statistical analysis

All statistical analyses were implemented in R (R Development Core Team, 2013) using the ANOVA function based on arcsine square root transformed proportions of settled larvae. For all models, two factors were included: cue sample nested within cue origin (Pacific/ Caribbean) and coral species. Significance of factors was evaluated using likelihood ratio tests (LRT). If a factor was found to be significant, a post-hoc Tukey’s HSD test was used to evaluate the significance of each pair-wise comparison. All assumptions of parametric testing were validated using diagnostic plots in R.

To visualize coral species-specific cue preferences, both principal components analysis (PCA) and non-metric multidimensional scaling (NMDS) ordination were used. PCA was computed using the cmdscale (R Development Core Team, 2013) and vegan (Jari Oksanen et al., 2013) packages. Bray-Curtis similarity coefficients were used for NMDS analysis using vegan package (Jari Oksanen et al., 2013). The resulting PCA and NMDS scores were visualized in two-dimensional ordination space.

Results

Caribbean spawn I

Larvae of the only coral species that was obtained, Montastraea franksi, exhibited distinct preferences for specific cues in the panel tested (Table 3, PLRT < 0.001). Settlement was significantly higher in response to Caribbean cues, although the cue from Pohnpei was only significantly surpassed by the most preferred Caribbean cue (Florida, FF) (Fig. 1A; Tukey’s HSD, p = 0.006). No recruits were observed in the control wells.

Table 2 Characteristics of the two most abundant operational taxonomic units (OTUs) in each cue sample including: the OTU name, length of the consensus sequence, percent of the mapped reads that mapped to that OTU, the best NCBI Blast hit for that OTU, if that blast hit was a CCA species, if that blast hit was in the phylum Rhodophyta, and the Genbank Accession Number for that reference OTU.

OTU	Genbank accession number	Length (bp)	% mapped reads	NCBI blast hit	CCA	Rhodophyta	
Aus1_1	KJ609529	498	54.2	Uncultured fungus	N	N	
Aus1_2	KJ609530	482	14.1	Uncultured fungus	N	N	
Aus2_1	KJ609525	514	36.9	Mastophoroideae	Y	Y	
Aus2_2	KJ609526	513	6.6	Mastophoroideae	Y	Y	
Bonaire_1	KJ609527	528	27.8	Order Gigartinales	N	Y	
Bonaire_2	KJ609528	516	15.4	Hydrolithion spp	Y	Y	
Florida_1	KJ609523	519	52.0	Subfamily Melobesioideae	Y	Y	
Florida_2	KJ609524	519	12.7	Subfamily Melobesioideae	Y	Y	
FGB_1	KJ609531	531	27.4	Order Corallinales	Y	Y	
FGB_2	KJ609532	520	21.6	Subfamily Melobesioideae	Y	Y	
Guam_1	KJ609521	520	26.3	Hydrolithon onkodes	Y	Y	
Guam_2	KJ609522	520	13.1	Hydrolithon onkodes	Y	Y	

Table 3 Summary statistics for settlement responsiveness for all Caribbean and Indo-Pacific species.

Likelihood ratio test (LRT) and Tukey’s HSD statistics for significant model terms testing the proportion of settlement in response to different CCA cues.

Experiment	Test	Factor	df	SS	F	p	
Caribbean Spawn I							
M. franksi	LRT	Cue	5	1.99	18.34	<0.001	
		Residuals	18	0.40	0.02		
	Tukey HSD	B–A1				<0.001	
		FF–A1				<0.001	
		FGB–A1				<0.001	
		P–A1				0.02	
		G–B				0.002	
		G–FF				<0.001	
		P–FF				0.007	
		G–FGB				0.003	
Pacific Spawn I	LRT	Cue	6	7.89	1.31	<0.001	
		Species	2	3.28	1.64	0.012	
		Cue ∗ Species	12	2.24	0.19	0.005	
		Residuals	104	7.52	0.07		
	Tukey HSD	Species					
		Mil–Liz				<0.001	
		Ten–Liz				<0.001	
		Cue					
		A2–A1				<0.001	
		B–A2				<0.001	
		FF–A2				<0.001	
		FGB–A2				<0.001	
		G–A2				<0.001	
		P–A2				<0.001	
		FF–B				0.015	
		P–B				0.027	
		G–FF				0.002	
		P–G				0.003	
		Cue ∗ Species					
		Favia Lizardensis					
		None					
		Acropora millepora					
		A2–A1				<0.001	
		A2–B				<0.001	
		A2–FF				0.011	
		A2–FGB				<0.001	
		A2–G				<0.001	
		A2–P				<0.001	
		Acropora tenuis					
		A2–A1				0.006	
		A2–B				0.004	
		A2–FGB				<0.001	
		A2–G				<0.001	
		FF–FGB				0.05	
		FF–G				0.03	
Caribbean Spawn II	LRT	Cue	6	2.17	0.36	<0.001	
		Species	1	2.44	2.44	<0.001	
		Cue ∗ Species	6	0.55	0.09	0.004	
		Residuals	70	2445.07			
	Tukey HSD	Species					
		Fra–Sti				<0.001	
		Cue					
		A2–A1				<0.001	
		B–A1				0.045	
		FF–A1				<0.001	
		FGB–A1				<0.001	
		A2–B				0.001	
		A2–G				<0.001	
		A2–P				<0.001	
		FF–G				<0.001	
		FGB–G				0.003	
		Cue ∗ Species					
		Diploria strigosa					
		A2–A1				0.002	
		A2–G				0.017	
		A2–P				0.018	
		B–A1				0.010	
		Montastraea franksi					
		A2–A1				<0.001	
		A2–B				<0.001	
		A2–G				<0.001	
		A2–P				0.05	
		FF–A1				0.004	
		FF–B				0.014	
		FF–G				0.004	
Notes.

Cues

A1 Australia 1

A2 Australia 2

B Bonaire

G Guam

FF Florida

FGB Flower Garden Banks

P Pohnpei

Species

Fra Montastraea franksi

Liz Favia lizardensis

Mil Acropora millepora

Str Diploria strigosa

Ten Acropora tenuis

Pacific spawn I

Both main effects of cue (PLRT < 0.001) and coral species (PLRT < 0.001) were significant, as well as their interaction (PLRT = 0.005), the latter indicating that the coral species differed significantly in their cue preferences (Fig. 2). There were no observable tendencies of Indo-Pacific larvae to prefer cues from either Indo-Pacific or Caribbean. Pairwise comparisons between species in their responses to settlement cues determined that both A. millepora and A. tenuis were different from F. lizardensis, but no significant difference was observed between these two acroporids (Tukey’s HSD, p = 0.483) (Table 3). With the exception of Ctenactis echinata that failed to respond to any cue, all species exhibited high response to the Australia 2 (A2) cue and also responded to Florida (FF) and Pohnpei (P) cues greater than those cues from Bonaire (B) and Guam (G) (Table 3). F. lizardensis responded to all cues; the only suggestion of specificity was a marginal, but insignificant, difference (Tukey’s HSD, p = 0.063) between A2 (70% settlement) and G (30% settlement). The acroporids were similar in their cue preferences, although A. tenuis settled in greater than A. millepora and demonstrated no selectivity between Australia 2 (A2) and Florida (FF) or Pohnpei (P). A. tenuis also preferred Florida (FF) cue over the Flower Garden Banks (FGB) (Tukey’s HSD, p = 0.05) and Bonaire (B) (Tukey’s HSD, p = 0.03) cues. No larvae of any species tested were observed to settle in control conditions.

Figure 2 Settlement responses of Pacific corals from Orpheus Island, GBR in 2010 (mean ± SE). Darker bars correspond to Caribbean cues, lighter bars to Pacific cues.

Caribbean spawn II

Similarly to the results of the Pacific spawn, there were significant main effects of cue (PLRT < 0.001) and species (PLRT < 0.001) and a significant interaction term (PLRT = 0.004) (Fig. 3, Table 3). The most preferred cue of D. strigosa was Australia 2 (A2), followed by all Caribbean cues. The tendency of M. franksi larvae to prefer Caribbean cues observed in 2010 was not detected in 2011, as M. franksi preferred A2 (which was not included in the 2010 panel) to any other cue in the panel. Compared to M. franksi, D. strigosa settled at a higher rate, regardless of cue (Tukey’s HSD, p < 0.001). No settlement was observed for the gonochoristic broadcaster Stephanocoenia intersepta regardless of the cue offered. No M. franksi larvae were observed to settle in the control conditions, however; for D. strigosa, an average of 3% of larvae spontaneously settled in control conditions (data not shown).

Figure 3 Settlement responses of Caribbean corals from the Flower Garden Banks in 2011 (mean ± SE). Darker bars correspond to Caribbean cues, lighter bars to Pacific cues.

Metabarcoding of cue samples

From the total 20,872 reads, 18,862 were left after quality filtering (∼90%). 15,217 reads mapped to the OTUs derived from 97% similarity clusters containing >1% of the total reads. Mapping efficiencies for each cue sample back to its OTUs was 66–98% with a mean of 81%. Rarefaction analysis indicated that our sequencing coverage efficiently captured sequence diversity in each sequenced sample (Fig. S1). The relative proportions of each taxonomic order differed between cue samples (Fig. 4). Australia 2 (A2), Florida (FF), Guam (G) and Flower Garden Banks (FGB) all contained >50% of the order Corallinales, to which crustose coralline algae (CCA) belong. Both Bonaire (B) and Guam (G) also contained high proportions (>25%) of filamentous red algal orders within the Phylum Rhodophyta (Gelidiales, Gigartinales and Peyssonneliales) (Fig. 4A) Interestingly Australia 1 (A1) contained no Corallinales reads and the majority of its OTUs remained taxonomically unplaced. NMDS also demonstrated the differences between cue communities showing cues with similar proportions of order Corallinales clustering more closely (Fig. 4B).

Figure 4 CCA cue community compositions.

(A) Relative proportions of mapped reads belonging to various taxonomic groups. (B) Non-metric multidimensional scaling (Bray-Curtis nMDS−2 dimensional) based on proportions of taxa in the cue communities.

The neighbor-joining tree constructed using the two most highly represented OTUs from each cue sample was well resolved, with bootstrap scores ranging from 0.54 to 1 (Fig. 5). Analysis of sequence similarity using BLAST confirmed that all but one (A1) of the successfully sequenced cues predominantly contained Rhodophyta (red algae) sequences. Of these, all but one OTU from Bonaire were from order Corallinales (CCAs). The two main clades in the neighbor-joining tree corresponded to the subfamilies Mastophorideae and Melobesioideae. One of the references from FGB was identified to the order Corallinales, but its family remained unresolved.

Figure 5 Neighbor-joining (NJ) tree of the two most abundant OTUs in each cue sample. Bootstrap support is shown at each node. Symbol (∗) indicates that the reference sequence belongs to order Corallinales, (∼) belongs to the Phylum Rhodophyta and (#) indicates that the taxonomic affiliation of the OTU could not be resolved.

Coral species-specific preferences

Both PCA and NMDS analyses demonstrated that corals exhibit species-specific cue preferences, with the exception of the two Acropora species that were similar to each other (Fig. 6). NMDS was superior to PCA at resolving these differences with a low stress value (0.0692) (Fig. 6B). For the PCA (Fig. 6A), component 1 (PCA1) explained 45% of the variation and component 2 (PCA2) explained 15%.

Figure 6 CCA cue preference differences.

Cue preferences differ between coral species from the Caribbean and Pacific (see legend), based on proportion of larvae that settled in response to the cue. (A) Principle component analysis (PCA) (B) Non-metric multidimensional scaling (Bray-Curtis nMDS, 2-dimensional).

Discussion

Caribbean larvae, with the exception of the gonochoric broadcaster S. intersepta that failed to respond to any cue, responded to the settlement cues tested in a similar manner to Pacific larvae, suggesting that the lack of recruitment observed in the Caribbean is not due to poor ability of larvae to perceive settlement cue. Furthermore, the panel of Caribbean cues tested here were very successful in inducing settlement of both Caribbean and Indo-Pacific corals tested (Figs. 1–3), demonstrating that effective cues are present on Caribbean reefs and were represented within our collection of cue samples. Previous studies of coral settlement, from both the Caribbean and Indo-Pacific, have demonstrated that coral larvae settle higher in response to certain species of CCAs over others (Harrington et al., 2004; Arnold, Steneck & Mumby, 2010; Price, 2010; Ritson-Williams et al., 2010). Our data confirm these results and further demonstrate that these preferences can vary substantially among broadcast-spawning coral species, even if these corals are from the same reef environment at the same location. In addition, some species, such as F. lizardensis, appear to be less specific overall and settle in high proportions regardless of cue type (at least for the cue panel tested here), while others did not respond to any cues tested (C. echinata, S. intersepta).

Preferences of Caribbean corals

Data from the pilot study in the Caribbean (2010) suggested the potential for co-adaptation between larval cue receptors and Caribbean cues, as the larvae of M. franksi settled in higher proportions in response to Caribbean cues rather than Pacific cues (Fig. 1). However, results of the second Caribbean spawning season (2011) did not support this hypothesis since both M. franksi and D. strigosa responded best to the newly introduced Pacific cue (A2). Beyond A2, Caribbean larvae settled well in response to Caribbean cues and even (in case of D. strigosa) tended to prefer them (Fig. 3), indicating that the Caribbean corals tested were fully capable of settlement in response to local Caribbean cues. M. franksi and D. strigosa also demonstrated species-specific cue preferences (Fig. 6). Year-to-year variation in settlement success for M. franksi was observed, with settlement in 2011 being less successful than 2010 (Figs. 1 and 3). Although great care was taken to culture larvae in identical conditions, unknown year-to-year variations in culture conditions may have influenced larval settlement. All cues were kept frozen, however each cue was collected at different times so settlement cue age may have altered their effectiveness through time by modifying cue stability. Therefore, the coral responses to the cues were only compared among coral species within the same field season. It is also possible that the year-to-year variation observed in this study reflects the natural stochasticity of the recruitment process or genetic difference between larval cohorts (Meyer et al., 2009).

Preferences of Pacific corals

No Indo-Pacific-wide trends were ever observed for the corals and cues tested here, but clear differences in cue preferences between coral species were apparent, with the two Acropora species exhibiting more specific settlement behavior (Figs. 2 and 6). The strict preferences of A. millepora and A. tenuis larvae have been reported previously (Harrington et al., 2004), and the similarity of their cue preferences observed in our experiments (Fig. 6) might be attributable to their phylogenetic proximity. Favia lizardensis was much less selective and high settlement rates were observed in response to most cues (Fig. 2). This result is similar to observations from its Caribbean congener, Favia fragum, which had previously been shown to be relatively indiscriminate in its settlement behavior (Nugues & Szmant, 2006), although it must be noted that F. fragum is a brooding rather than broadcast-spawning species. While our data do not formally allow drawing taxonomy-related conclusions, the similarity of cue preferences in congeneric coral species across our cue panel is notable and might reflect the general pattern of cue preference evolution.

Corals that would not settle: Ctenactis echinata and Stephanocoenia intersepta

Both species demonstrated complete lack of settlement response to the same cue panel that successfully induced metamorphosis in other corals, and therefore these species represent the most extreme demonstration of divergent cue preferences among the corals tested. While C. echinata was only tested at five days post fertilization, leaving open a possibility that the culture had not yet reached competency, S. intersepta was assayed for settlement for approximately two months and was still never observed to settle for any cue. Interestingly, these species are from different oceans but share one key life history trait: they are both gonochoric (i.e., have separate sexes) whereas all other coral species tested were hermaphroditic. It is tempting to speculate that this shared life history trait underlies their lack of response in our settlement trials. Previous work on a gonorchoric, broadcast-spawning gorgonian coral demonstrated that adult proximity to conspecifics had a large effect on reproductive success (Coffroth & Lasker, 1998), one of the possibilities being that gonochoric corals might need additional cues from conspecifics to ensure close proximity and efficient fertilization during spawning (Tamburri, Zimmer & Zimmer, 2007). While we cannot discount that these corals were unresponsive because they had not reached competence or they were not offered appropriate cues, we believe that this hypothesis merits detailed investigation in the future.

Composition of the cue communities

Each cue community differed in its relative proportions of taxa; however, most cues that were effective at inducing settlement in the corals tested here contained >50% order Corallinales, the order which contains CCAs (Fig. 4). Notably, one cue (A1) yielded no Corallinales reads yet still induced settlement, although it was among the least effective. Two major CCA sub-families were represented in the cue communities: Mastophorideae and Melobesioideae (Fig. 5). These taxonomic groups have previously been shown to be strong larval settlement inducers (Heyward & Negri, 1999; Harrington et al., 2004; Ritson-Williams et al., 2010), indicating that our cue collections efforts were, in fact, at least taxonomically-related to previously established settlement cues for corals. While we could only discriminate taxa to the order or family level, this is the first study to create a sequence database of natural coral settlement cues.

Possible consequences of coral species-specific cue preferences

Settlement choice has been shown to strongly influence post-settlement survival, illustrating the consequences of larval selectivity (Babcock & Mundy, 1996; Harrington et al., 2004). Divergent larval settlement preferences correlating with cue availability in the adults’ natural habitat have been previously demonstrated for two coral species from Guam, Stylaraea punctata and Goniastrea retiformis (Golbuu & Richmond, 2007). However, divergent preferences between these species were expected since they do not co-occur in the same reef environment; moreover, S. punctata is a brooder while G.retiformis is a broadcast spawner. Our study is the first to document species-specific preferences in a panel of settlement cues among broadcast-spawning corals from the same reef community for both the Indo-Pacific and the Caribbean (Fig. 6), and it is tempting to speculate that these preferences might play a role in coral community assembly. While our study did not, by any means, exhaust all potential cues available for corals arriving to reefs, it did demonstrate that some coral species are considerably more “choosy”. This finding is especially concerning given ongoing climate change, since CCA are among the most sensitive reef organisms to both warming and acidification (Webster et al., 2011; Ragazzola et al., 2012; Doropoulos & Diaz-Pulido, 2013; Webster et al., 2013). Diminishing CCA abundances and effectiveness as settlement inducers might be accompanied by a reduction in CCA diversity, which in turn could lead to coral community shifts in favor of less selective coral species that do not require particular settlement cues.

Our research demonstrates that Caribbean coral larvae can respond to the local settlement cues on par with Indo-Pacific larvae, suggesting that, at least in the lab, interactions between corals and cues on Caribbean reefs have not been compromised relative to the Indo-Pacific. However, it is clear that other processes are causing region-wide Caribbean recruitment failure, and identifying these processes should remain a research priority.

Supplemental Information

Figure S1 Cue rarefaction analysis

Rarefaction analysis of sequence coverage. Average number of OTUs identified in each cue sample at various coverage depths (number of reads).

Click here for additional data file.

Supplemental Information 1 Rarefaction analysis and results

SI_cca_rarefaction.pl: Perl script that completes rarefaction analysis of sequence data S2_rarefaction_figs.R: Legend: R script that plots output of rarefaction analysis of sequence data.

Click here for additional data file.

We acknowledge personnel at the FGBNMS (E Hickerson & GP Schmahl) and Orpheus Island Research Station for permits (FGB: FGBNMS-2009-005-A2, A3 and GBR: G10/33943.1) and the FGBNMS for boat time. We would also like to thank the reviewers and the editor for very detailed and useful revisions on the manuscript.

Additional Information and Declarations

Competing Interests

Author Contributions

Field Study Permissions

DNA Deposition

The authors declare there are no competing interests.

Sarah W. Davies conceived and designed the experiments, performed the experiments, analyzed the data, contributed reagents/materials/analysis tools, wrote the paper, prepared figures and/or tables, reviewed drafts of the paper.

Eli Meyer conceived and designed the experiments, performed the experiments, reviewed drafts of the paper.

Sarah M. Guermond performed the experiments, reviewed drafts of the paper.

Mikhail V. Matz conceived and designed the experiments, performed the experiments, contributed reagents/materials/analysis tools, reviewed drafts of the paper.

The following information was supplied relating to ethical approvals (i.e., approving body and any reference numbers):

Flower Garden Banks National Marine Sanctuary: FGBNMS-2009-005-A2, A3

Orpheus Island Research Station: G10/33943.1

The following information was supplied regarding the deposition of DNA sequences:

Top OTU references have been deposited on Genbank: Accession numbers KJ609521–KJ609532.

In addition, raw sff can be found on the Sequence Read Archive (SRA) Accession Number: SRP040596.

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
