# Peer review of "A cross-ocean comparison of responses to settlement cues in reef-building corals"

_PeerJ, doi:10.7717/peerj.333_

## Round 0.1 · original submission · Major Revisions

The ms has been reviewed by three reviewers and the recommendations varied substantially. After going through the comments and ms, I see the points of the first reviewer. Please go through the comments carefully and if you decide to resubmit your revised ms, please provide detailed responses to each major issue raised by the reviewers. Your ms will be reviewed again.

Thanks. PY

Reviewer 1 ·

Basic reporting

no comments

Experimental design

The methods these authors use are not rigorous enough to publish this data. One of the most basic principles of experimental science is that experiments could be repeated by other researchers: however, by picking up random piece of rubble from different locations in each ocean there is no systematic method of determining settlement substrata that these larvae were exposed to, making these experiments impossible to repeat. Recent literature has shown that some coral larvae settle in response to different species of crustose coralline algae, however, these authors did not determine the species they used in their experiments. Instead of identifying the coralline algae the authors sequenced them, which led to family level (if that) identification. This is not adequate to understand which species of coralline algae the larvae are responding to. This research is not repeatable, there is no way for another research to find the same species of coralline algae that were used in these experiments. A better method would be to choose specific species of CCA to test settlement specificity among coral species from different oceans.

In addition, there was no attempt to characterize an ocean using the settlement substrata. How can the authors determine if there is a “region-wide disruptions of the settlement process” if they haven’t characterized the region. By picking up random pieces of rubble the authors could have selected very rare species or extremely common species at each location: however, we will never know which because the authors only have 3 or 4 samples from each ocean. Indeed each of these samples are from very different locations within the ocean, for instance the samples from the Caribbean were collected from Florida, Bonaire, and the Flower Garden Banks. These are three different types of reefs, were the samples collect from the same depths? The same types of habitats? The same environmental conditions? Obviously all of these parameters are variable and without multiple collections at each location it would be impossible to determine what is “average” substratum at each location, let alone each ocean basin. By using 2 or 3 species that are dominant at multiple sites in the Caribbean and Pacific Ocean you could compare coral settlement specificity to the most abundant CCA species in each ocean.

Your methods don’t have a settlement substrata, I know that other authors have also done settlement in well plates, but surely you can see that larval settlement on plastic isn’t ecologically relevant. Take the time to identify the settlement substrata. If you are interested in coralline algae, identify it to species so that other researchers can repeat or build on your research. If you are interested in biofilms, use your NGS techniques to determine the biofilm communities. Right now you haven’t characterized either.

Validity of the findings

It is also unclear to me how reliable your results are. At least for M. franksi, you got very different results in each year. This suggests that if you had repeated your experiments for the other corals you might also get different results. This makes me suspicious that the results aren’t trustworthy, another reason they shouldn’t be published.

Additional comments

I understand how much work it is to do these larval experiments. However, I would strongly recommend that the authors reconsider the experiments they have presented in this manuscript. I don’t see how these results help us understand coral larval settlement. Overall I can not recommend this research to be published. While I think you have an interesting question and hypothesis your methods do not adequately address your question and your results appear to be unreliable.

Reviewer 2 ·

Basic reporting

No Comments

Experimental design

Different settlement of larvae might due to low competency of larvae. It would be nice to have a positive control (chemical inducer of larval settlement) to test this hypothesis.

Validity of the findings

Results are reproducible and of the high scientific standard. Conclusions are well justified.

Additional comments

In this manuscript the authors tried to understand the reason of low resilience of Caribbean reefs in comparison to Indo-Pacific ones. They proposed that this is due to the low recruitment of corals in Caribbean because of the absence of settlement cues or low larval ability to respond for these cues. In order to test this, the authors performed settlement experiments with three species of broadcast spawning Caribbean corals (Montastraea franksi, Diploria strigosa and Stephanocoenia intersepta) and four Indo-Pacific corals (Acropora millepora, A. tenuis, Favia lizardensis and Ctenactis echinata) using crustose coralline algae (CCA) from Indo-Pacific and Caribbean. CCA communities were characterized using metagenomic analysis of their 18S rRNA. The authors found that CCA cues and species of corals had a significant impact on larval settlement. Larvae of 2 species of corals (C. echinata and S. intersepta) failed to respond to any cue. Overall, the results suggested that poor coral recruitment in Caribbean is not due to absence of cues or low larval response but driven by other factors. Since this study was performed in the laboratory, these results cannot be fully applied to the field conditions. Additionally, it is possible that other cues rather than CCA could be involved in larval settlement. The manuscript is well written. Results are reproducible and of the high scientific standard. Conclusions are well justified. Overall, I recommend publication of this manuscript after a minor revision. My comments are below:
P.5 line 73. Change to 80o C
Figures 1a, 2, 3. Include statistical results.
Table 2. Why did you analyze only 2 most abundant OTUs? It is important to reveal identity of those OTUs that cause dissimilarities between CCA communities.
Present a rarefaction curve for your metagenomic analysis.

·

Basic reporting

no issues with this category. A sound paper!

Experimental design

The results of this study hinge on the ability to compare CCa samples collected at different times, locations and stored over various periods of time. While the authors acknowledge this point, it would have been good to see a verification of cue stability (inductive potential) by comparing stored vs fresh cues. If the authors have data that compare old and new cues of the same CCA tested in 2010 and 2011, this would strengthen the manuscript.
Please check l.73. Do you mean minus 80 degrees?

Validity of the findings

all fine

Additional comments

Nice and timely study. Just few comments:

Title/abstract. since this is a lab-based study, this needs to be made clear somewhere in Title and/or abstract
l. 226: for clarity maybe say "coral species-specific"
l. 256: please elaborate on cue stability

---

## Round 0.2 · accepted · Accept

Dear authors, Your revised ms has been further assessed by the reviewers and they are happy with your revision. I concur their recommendation and thus, am happy to inform you that your ms is officially accepted for publication in PeerJ.

Reviewer 2 ·

Basic reporting

I am satisfied with all changes

Experimental design

I have no comments

Validity of the findings

The findings are important and valid.

Additional comments

I am satisfied with all changes

·

Basic reporting

I believe the ms is ready for publication.

Experimental design

ok

Validity of the findings

ok

Additional comments

The revised ms now warrants publication in my view.